# FixMatch: Simplifying Semi-Supervised Learning with Consistency and Confidence

**Kihyuk Sohn**[*]   **David Berthelot**[*]   **Chun-Liang Li**   **Zizhao Zhang**   **Nicholas Carlini**
**Ekin D. Cubuk**   **Alex Kurakin**   **Han Zhang**   **Colin Raffel**
Google Research
{kihyuks,dberth,chunliang,zizhaoz,ncarlini,
cubuk,kurakin,zhanghan,craffel}@google.com

## Abstract

Semi-supervised learning (SSL) provides an effective means of leveraging unlabeled data to improve a model's performance. This domain has seen fast progress recently, at the cost of requiring more complex methods. In this paper we propose FixMatch, an algorithm that is a significant simplification of existing SSL methods. FixMatch first generates pseudo-labels using the model's predictions on weakly-augmented unlabeled images. For a given image, the pseudo-label is only retained if the model produces a high-confidence prediction. The model is then trained to predict the pseudo-label when fed a strongly-augmented version of the same image. Despite its simplicity, we show that FixMatch achieves state-of-the-art performance across a variety of standard semi-supervised learning benchmarks, including 94.93% accuracy on CIFAR-10 with 250 labels and 88.61% accuracy with 40 – just 4 labels per class. We carry out an extensive ablation study to tease apart the experimental factors that are most important to FixMatch's success. The code is available at https://github.com/google-research/fixmatch.

## 1   Introduction

Deep neural networks have become the de facto model for computer vision applications. Their success is partially attributable to their *scalability*, i.e., the empirical observation that training them on larger datasets produces better performance [30, 20, 42, 55, 41, 21]. Deep networks often achieve their strong performance through supervised learning, which requires a labeled dataset. The performance benefit conferred by the use of a larger dataset can therefore come at a significant cost since labeling data often requires human labor. This cost can be particularly extreme when labeling must be done by an expert (for example, a doctor in medical applications).

A powerful approach for training models on a large amount of data without requiring a large amount of labels is *semi-supervised learning* (SSL). SSL mitigates the requirement for labeled data by providing a means of leveraging unlabeled data. Since unlabeled data can often be obtained with minimal human labor, any performance boost conferred by SSL often comes with low cost. This has led to a plethora of SSL methods that are designed for deep networks [33, 46, 24, 51, 4, 54, 3, 25, 45, 52].

A popular class of SSL methods can be viewed as producing an artificial label for unlabeled images and training the model to predict the artificial label when fed unlabeled images as input. For example, pseudo-labeling [25] (also called self-training [32, 55, 44, 47]) uses the model's class prediction as a label to train against. Similarly, consistency regularization [2, 46, 24] obtains an artificial label using the model's predicted distribution after randomly modifying the input or model function.

---

[*]Equal contribution.

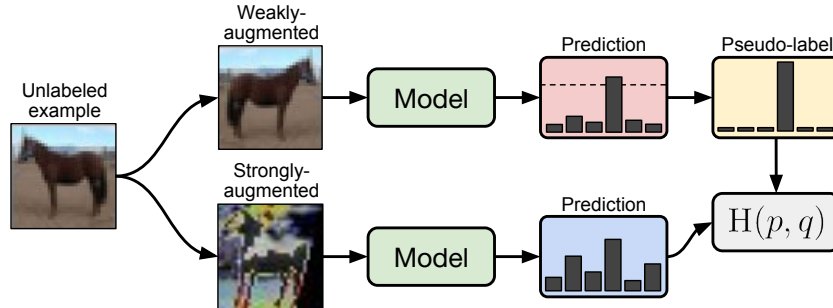

Figure 1: Diagram of FixMatch. A weakly-augmented image (top) is fed into the model to obtain predictions (red box). When the model assigns a probability to any class which is above a threshold (dotted line), the prediction is converted to a one-hot pseudo-label. Then, we compute the model's prediction for a strong augmentation of the same image (bottom). The model is trained to make its prediction on the strongly-augmented version match the pseudo-label via a cross-entropy loss.

In this work, we break the trend of recent state-of-the-art methods that combine increasingly complex mechanisms [4, 54, 3] and produce a method that is simpler, but also more accurate. Our algorithm, FixMatch, produces artificial labels using both consistency regularization and pseudo-labeling. Crucially, the artificial label is produced based on a *weakly*-augmented unlabeled image (e.g., using only flip-and-shift data augmentation) which is used as a target when the model is fed a *strongly*-augmented version of the same image. Inspired by UDA [54] and ReMixMatch [3], we leverage Cutout [14], CTAugment [3], and RandAugment [11] for strong augmentation, which all produce heavily-distorted versions of a given image. Following the approach of pseudo-labeling [25], we only retain an artificial label if the model assigns a high probability to one of the possible classes. A diagram of FixMatch is shown in fig. 1.

Despite its simplicity, we show that *FixMatch obtains state-of-the-art performance on the most commonly-studied SSL benchmarks*. For example, FixMatch achieves $94.93\%$ accuracy on CIFAR-10 with 250 labeled examples compared to the previous state-of-the-art of $93.73\%$ [3] in the standard experimental setting from [36]. We also explore the limits of our approach by applying it in the extremely-scarce-labels regime, obtaining $88.61\%$ accuracy on CIFAR-10 with only 4 labels per class. Since FixMatch is a simplification of existing approaches but achieves substantially better performance, we include an extensive ablation study to determine which factors contribute the most to its success. A key benefit of FixMatch being a simplification of existing methods is that it requires many fewer additional hyperparameters. As such, it allows us to perform an extensive ablation study of each of them. Our ablation study also includes basic fully-supervised learning experimental choices that are often ignored or not reported when new SSL methods are proposed (such as the optimizer or learning rate schedule).

## 2 FixMatch

FixMatch is a combination of two approaches to SSL: Consistency regularization and pseudo-labeling. Its main novelty comes from the combination of these two ingredients as well as the use of a separate weak and strong augmentation when performing consistency regularization. In this section, we first review consistency regularization and pseudo-labeling before describing FixMatch in detail. We also describe the other factors, such as regularization, which contribute to FixMatch's empirical success.

For an $L$-class classification problem, let $\mathcal{X} = \big\{(x_b, p_b) : b \in (1, \ldots, B)\big\}$ be a batch of $B$ labeled examples, where $x_b$ are the training examples and $p_b$ are one-hot labels. Let $\mathcal{U} = \big\{u_b : b \in (1, \ldots, \mu B)\big\}$ be a batch of $\mu B$ unlabeled examples where $\mu$ is a hyperparameter that determines the relative sizes of $\mathcal{X}$ and $\mathcal{U}$. Let $p_{\mathrm{m}}(y \mid x)$ be the predicted class distribution produced by the model for input $x$. We denote the cross-entropy between two probability distributions $p$ and $q$ as $\mathrm{H}(p, q)$. We perform two types of augmentations as part of FixMatch: strong and weak, denoted by $\mathcal{A}(\cdot)$ and $\alpha(\cdot)$ respectively. We describe the form of augmentation we use for $\mathcal{A}$ and $\alpha$ in section 2.3.

## 2.1 Background

*Consistency regularization* is an important component of recent state-of-the-art SSL algorithms. Consistency regularization utilizes unlabeled data by relying on the assumption that the model should output similar predictions when fed perturbed versions of the same image. This idea was first proposed in [2] and popularized by [46, 24], where the model is trained both via a standard supervised classification loss and on unlabeled data via the loss function

$$\sum_{b=1}^{\mu B} \| p_{\mathrm{m}}(y \,|\, \alpha(u_b)) - p_{\mathrm{m}}(y \,|\, \alpha(u_b)) \|_2^2 \qquad (1)$$

Note that both $\alpha$ and $p_{\mathrm{m}}$ are stochastic functions, so the two terms in eq. (1) will indeed have different values. Extensions to this idea include using an adversarial transformation in place of $\alpha$ [33], using a running average or past model predictions for one invocation of $p_{\mathrm{m}}$ [51, 24], using a cross-entropy loss in place of the squared $\ell^2$ loss [33, 54, 3], using stronger forms of augmentation [54, 3], and using consistency regularization as a component in a larger SSL pipeline [4, 3].

*Pseudo-labeling* leverages the idea of using the model itself to obtain artificial labels for unlabeled data [32, 47]. Specifically, this refers to the use of "hard" labels (i.e., the $\arg\max$ of the model's output) and only retaining artificial labels whose largest class probability fall above a predefined threshold [25]. Letting $q_b = p_{\mathrm{m}}(y|u_b)$, pseudo-labeling uses the following loss function:

$$\frac{1}{\mu B} \sum_{b=1}^{\mu B} \mathbb{1}(\max(q_b) \geq \tau) \, \mathrm{H}(\hat{q}_b, q_b) \qquad (2)$$

where $\hat{q}_b = \arg\max(q_b)$ and $\tau$ is the threshold. For simplicity, we assume that $\arg\max$ applied to a probability distribution produces a valid "one-hot" probability distribution. The use of a hard label makes pseudo-labeling closely related to entropy minimization [17, 45], where the model's predictions are encouraged to be low-entropy (i.e., high-confidence) on unlabeled data.

## 2.2 Our Algorithm: FixMatch

The loss function for FixMatch consists of two cross-entropy loss terms: a supervised loss $\ell_s$ applied to labeled data and an unsupervised loss $\ell_u$. Specifically, $\ell_s$ is just the standard cross-entropy loss on weakly augmented labeled examples:

$$\ell_s = \frac{1}{B} \sum_{b=1}^{B} \mathrm{H}(p_b, p_{\mathrm{m}}(y \mid \alpha(x_b))) \qquad (3)$$

FixMatch computes an artificial label for each unlabeled example[2] which is then used in a standard cross-entropy loss. To obtain an artificial label, we first compute the model's predicted class distribution given a *weakly*-augmented version of a given unlabeled image: $q_b = p_{\mathrm{m}}(y \mid \alpha(u_b))$. Then, we use $\hat{q}_b = \arg\max(q_b)$ as a pseudo-label, except we enforce the cross-entropy loss against the model's output for a *strongly*-augmented version of $u_b$:

$$\ell_u = \frac{1}{\mu B} \sum_{b=1}^{\mu B} \mathbb{1}(\max(q_b) \geq \tau) \, \mathrm{H}(\hat{q}_b, p_{\mathrm{m}}(y \mid \mathcal{A}(u_b))) \qquad (4)$$

where $\tau$ is a scalar hyperparameter denoting the threshold above which we retain a pseudo-label. The loss minimized by FixMatch is simply $\ell_s + \lambda_u \ell_u$ where $\lambda_u$ is a fixed scalar hyperparameter denoting the relative weight of the unlabeled loss. We present a complete algorithm for FixMatch in algorithm 1 of the supplementary material.

While eq. (4) is similar to the pseudo-labeling loss in eq. (2), it is crucially different in that the artificial label is computed based on a weakly-augmented image and the loss is enforced against the model's output for a strongly-augmented image. This introduces a form of consistency regularization which, as we will show in section 5, is crucial to FixMatch's success. We also note that it is typical in modern SSL algorithms to increase the weight of the unlabeled loss term ($\lambda_u$) during training [51, 24, 4, 3, 36]. We found that this was unnecessary for FixMatch, which may be due to the fact

that $\max(q_b)$ is typically less than $\tau$ early in training. As training progresses, the model's predictions become more confident and it is more frequently the case that $\max(q_b) > \tau$. This suggests that pseudo-labeling may produce a natural curriculum "for free". Similar justifications have been used in the past for ignoring low-confidence predictions in visual domain adaptation [15].

### 2.3 Augmentation in FixMatch

FixMatch leverages two kinds of augmentations: "weak" and "strong". In all of our experiments, weak augmentation is a standard flip-and-shift augmentation strategy. Specifically, we randomly flip images horizontally with a probability of $50\%$ on all datasets except SVHN and we randomly translate images by up to $12.5\%$ vertically and horizontally.

For "strong" augmentation, we experiment with two methods based on AutoAugment [10], which are then followed by the Cutout [14]. AutoAugment uses reinforcement learning to find an augmentation strategy comprising transformations from the Python Imaging Library.[3] This requires labeled data to learn the augmentation strategy, making it problematic to use in SSL settings where limited labeled data is available. As a result, variants of AutoAugment which do not require the augmentation strategy to be learned ahead of time with labeled data, such as RandAugment [11] and CTAugment [3], have been proposed. Instead of using a learned strategy, both RandAugment and CTAugment randomly select transformations for each sample. For RandAugment, the magnitude that controls the severity of all distortions is randomly sampled from a pre-defined range (RandAugment with random magnitude was also used for UDA by [54]), whereas the magnitudes of individual transformations are learned on-the-fly for CTAugment. Refer to appendix E for more details.

### 2.4 Additional important factors

Semi-supervised performance can be substantially impacted by factors other than the SSL algorithm used because considerations like the amount of regularization can be particularly important in the low-label regime. This is compounded by the fact that the performance of deep networks trained for image classification can heavily depend on the architecture, optimizer, training schedule, etc. These factors are typically not emphasized when new SSL algorithms are introduced. Instead, we endeavor to quantify their importance and highlight which ones have a significant impact on performance. Most analysis is performed in section 5. In this section we identify a few key considerations.

First, as mentioned above, we find that regularization is particularly important. In all of our models and experiments, we use simple weight decay regularization. We also found that using the Adam optimizer [22] resulted in worse performance and instead use standard SGD with momentum [50, 40, 34]. We did not find a substantial difference between standard and Nesterov momentum. For a learning rate schedule, we use a cosine learning rate decay [28] which sets the learning rate to $\eta \cos\left(\frac{7\pi k}{16K}\right)$ where $\eta$ is the initial learning rate, $k$ is the current training step, and $K$ is the total number of training steps. Finally, we report final performance using an exponential moving average of model parameters.

### 2.5 Extensions of FixMatch

Due to its simplicity, FixMatch can be readily extended with techniques in SSL literature. For example, both Augmentation Anchoring (where $M$ strong augmentations are used for consistency regularization for each unlabeled example) and Distribution Alignment (which encourages the model predictions to have the same class distribution as the labeled set) from ReMixMatch [3] can be straightforwardly applied to FixMatch. Moreover, one may replace strong augmentations in FixMatch with modality-agnostic augmentation strategies, such as MixUp [59] or adversarial perturbations [33]. We present some exploration and experiments with these extensions in appendix D.

## 3 Related work

Semi-supervised learning is a mature field with a huge diversity of approaches. In this review, we focus on methods closely related to FixMatch. Broader introductions are provided in [60, 61, 6].

The idea behind self-training has been around for decades [47, 32]. The generality of self-training (i.e., using a model's predictions to obtain artificial labels for unlabeled data) has led it to be applied in many domains including NLP [31], object detection [44], image classification [25, 55], domain

| Algorithm | Artificial label augmentation | Prediction augmentation | Artificial label post-processing | Notes |
|---|---|---|---|---|
| TS / Π-Model | Weak | Weak | None | |
| Temporal Ensembling | Weak | Weak | None | Uses model from earlier in training |
| Mean Teacher | Weak | Weak | None | Uses an EMA of parameters |
| Virtual Adversarial Training | None | Adversarial | None | |
| UDA | Weak | Strong | Sharpening | Ignores low-confidence artificial labels |
| MixMatch | Weak | Weak | Sharpening | Averages multiple artificial labels |
| ReMixMatch | Weak | Strong | Sharpening | Sums losses for multiple predictions |
| FixMatch | Weak | Strong | Pseudo-labeling | |

Table 1: Comparison of SSL algorithms which include a form of consistency regularization and which (optionally) apply some form of post-processing to the artificial labels. We only mention those components of the SSL algorithm relevant to producing the artificial labels (for example, Virtual Adversarial Training additionally uses entropy minimization [17], MixMatch and ReMixMatch also use MixUp [59], UDA includes additional techniques like training signal annealing, etc.).

adaptation [62], to name a few. Pseudo-labeling refers to a specific variant where model predictions are converted to hard labels [25], which is often used along with a confidence-based thresholding that retains unlabeled examples only when the classifier is sufficiently confident (e.g., [44]). While some studies have suggested that pseudo-labeling is not competitive against other modern SSL algorithms on its own [36], recent SSL algorithms have used pseudo-labeling as a part of their pipeline to produce better results [1, 39]. As mentioned above, pseudo-labeling results in a form of entropy minimization [17] which has been used as a component for many SSL techniques [33].

Consistency regularization was first proposed by [2] and later referred to as "Transformation/Stability" (or TS for short) [46] or the "Π-Model" [43]. Early extensions included using an exponential moving average of model parameters [51] or using previous model checkpoints [24] when producing artificial labels. Several methods have been used to produce random perturbations including data augmentation [15], stochastic regularization (e.g. Dropout [49]) [46, 24], and adversarial perturbations [33]. More recently, it has been shown that using strong data augmentation can produce better results [54, 3]. These heavily-augmented examples are almost certainly outside of the data distribution, which has in fact been shown to be beneficial for SSL [12]. Noisy Student [55] has integrated these techniques into a self-training framework and demonstrated impressive performance on ImageNet with additional massive amount of unlabeled data.

Of the aforementioned work, FixMatch bears the closest resemblance to two recent methods: Unsupervised Data Augmentation (UDA) [54] and ReMixMatch [3]. They both use a weakly-augmented example to generate an artificial label and enforce consistency against strongly-augmented examples. Neither of them uses pseudo-labeling, but both approaches "sharpen" the artificial label to encourage the model to produce high-confidence predictions. UDA in particular also only enforces consistency when the highest probability in the predicted class distribution for the artificial label is above a threshold. The thresholded pseudo-labeling of FixMatch has a similar effect to sharpening. In addition, ReMixMatch anneals the weight of the unlabeled data loss, which we omit from FixMatch because we posit that the thresholding used in pseudo-labeling has a similar effect (as mentioned in section 2.2). These similarities suggest that FixMatch can be viewed as a substantially simplified version of UDA and ReMixMatch, where we have combined two common techniques (pseudo-labeling and consistency regularization) while removing many components (sharpening, training signal annealing from UDA, distribution alignment and the rotation loss from ReMixMatch, etc.).

Since the core of FixMatch is a simple combination of two existing techniques, it also bears substantial similarities to many previously-proposed SSL algorithms. We provide a concise comparison of each of these techniques in table 1 where we list the augmentation used for the artificial label, the model's prediction, and any post-processing applied to the artificial label. A more thorough comparison of these different algorithms and their constituent approaches is provided in the following section.

## 4 Experiments

We evaluate the efficacy of FixMatch on several SSL image classification benchmarks. Specifically, we perform experiments with varying amounts of labeled data and augmentation strategies on CIFAR-10/100 [23], SVHN [35], STL-10 [9], and ImageNet [13], following standard SSL evaluation protocols [36, 4, 3]. In many cases, we perform experiments with fewer labels than previously considered since FixMatch shows promise in extremely label-scarce settings. Note that we use an identical set of

| | CIFAR-10 | | | CIFAR-100 | | | SVHN | | | STL-10 |
|---|---|---|---|---|---|---|---|---|---|---|
| Method | 40 labels | 250 labels | 4000 labels | 400 labels | 2500 labels | 10000 labels | 40 labels | 250 labels | 1000 labels | 1000 labels |
| Π-Model | - | $54.26_{\pm3.97}$ | $14.01_{\pm0.38}$ | - | $57.25_{\pm0.48}$ | $37.88_{\pm0.11}$ | - | $18.96_{\pm1.92}$ | $7.54_{\pm0.36}$ | $26.23_{\pm0.82}$ |
| Pseudo-Labeling | - | $49.78_{\pm0.43}$ | $16.09_{\pm0.28}$ | - | $57.38_{\pm0.46}$ | $36.21_{\pm0.19}$ | - | $20.21_{\pm1.09}$ | $9.94_{\pm0.61}$ | $27.99_{\pm0.83}$ |
| Mean Teacher | - | $32.32_{\pm2.30}$ | $9.19_{\pm0.19}$ | - | $53.91_{\pm0.57}$ | $35.83_{\pm0.24}$ | - | $3.57_{\pm0.11}$ | $3.42_{\pm0.07}$ | $21.43_{\pm2.39}$ |
| MixMatch | $47.54_{\pm11.50}$ | $11.05_{\pm0.86}$ | $6.42_{\pm0.10}$ | $67.61_{\pm1.32}$ | $39.94_{\pm0.37}$ | $28.31_{\pm0.33}$ | $42.55_{\pm14.53}$ | $3.98_{\pm0.23}$ | $3.50_{\pm0.28}$ | $10.41_{\pm0.61}$ |
| UDA | $29.05_{\pm5.93}$ | $8.82_{\pm1.08}$ | $4.88_{\pm0.18}$ | $59.28_{\pm0.88}$ | $33.13_{\pm0.22}$ | $24.50_{\pm0.25}$ | $52.63_{\pm20.51}$ | $5.69_{\pm2.76}$ | $2.46_{\pm0.24}$ | $7.66_{\pm0.56}$ |
| ReMixMatch | $19.10_{\pm9.64}$ | $\mathbf{5.44}_{\pm0.05}$ | $4.72_{\pm0.13}$ | $\mathbf{44.28}_{\pm2.06}$ | $\mathbf{27.43}_{\pm0.31}$ | $\mathbf{23.03}_{\pm0.56}$ | $3.34_{\pm0.20}$ | $2.92_{\pm0.48}$ | $2.65_{\pm0.08}$ | $5.23_{\pm0.45}$ |
| FixMatch (RA) | $13.81_{\pm3.37}$ | $5.07_{\pm0.65}$ | $4.26_{\pm0.05}$ | $48.85_{\pm1.75}$ | $28.29_{\pm0.11}$ | $22.60_{\pm0.12}$ | $3.96_{\pm2.17}$ | $2.48_{\pm0.38}$ | $2.28_{\pm0.11}$ | $7.98_{\pm1.50}$ |
| FixMatch (CTA) | $\mathbf{11.39}_{\pm3.35}$ | $5.07_{\pm0.33}$ | $4.31_{\pm0.15}$ | $49.95_{\pm3.01}$ | $28.64_{\pm0.24}$ | $23.18_{\pm0.11}$ | $7.65_{\pm7.65}$ | $\mathbf{2.64}_{\pm0.64}$ | $2.36_{\pm0.19}$ | $\mathbf{5.17}_{\pm0.63}$ |

Table 2: Error rates for CIFAR-10, CIFAR-100, SVHN and STL-10 on 5 different folds. FixMatch (RA) uses RandAugment [11] and FixMatch (CTA) uses CTAugment [3] for strong-augmentation. All baseline models (Π-Model [43], Pseudo-Labeling [25], Mean Teacher [51], MixMatch [4], UDA [54], and ReMixMatch [3]) are tested using the same codebase.

hyperparameters ($\lambda_u = 1$, $\eta = 0.03$, $\beta = 0.9$, $\tau = 0.95$, $\mu = 7$, $B = 64$, $K = 2^{20}$)[4] across all amounts of labeled examples and datasets other than ImageNet. A complete list of hyperparameters is reported in appendix B.1. We include an extensive ablation study in section 5 to tease apart the importance of the different components and hyperparameters of FixMatch, including factors that are not explicitly part of the SSL algorithm such as the optimizer and learning rate.

## 4.1 CIFAR-10, CIFAR-100, and SVHN

We compare FixMatch to various existing methods on the standard CIFAR-10, CIFAR-100, and SVHN benchmarks. As suggested by [36], we reimplemented all existing baselines and performed all experiments using the same codebase. In particular, we use the same network architecture and training protocol, including the optimizer, learning rate schedule, data preprocessing, etc. across all SSL methods. Following [4], we used a Wide ResNet-28-2 [56] with 1.5M parameters for CIFAR-10 and SVHN, WRN-28-8 for CIFAR-100, and WRN-37-2 for STL-10. For baselines, we consider methods that are similar to FixMatch and/or are state-of-the-art: Π-Model [43], Mean Teacher [51], Pseudo-Label [25], MixMatch [4], UDA [54], and ReMixMatch [3]. Besides [3], previous work has not considered fewer than 25 labels per class on these benchmarks. Performing better with less supervision is the central goal of SSL in practice since it alleviates the need for labeled data. We also consider the setting where only 4 labeled images are given for each class on each dataset. As far as we are aware, we are the first to run *any* experiments at 4 labels per class on CIFAR-100.

We report the performance of all baselines along with FixMatch in table 2. We compute the mean and variance of accuracy when training on 5 different "folds" of labeled data. We omit results with 4 labels per class for Π-Model, Mean Teacher, and Pseudo-Labeling since the performance was poor at 250 labels. MixMatch, ReMixMatch, and UDA all perform reasonably well with 40 and 250 labels, but we find that FixMatch substantially outperforms each of these methods while nevertheless being simpler. For example, FixMatch achieves an average error rate of 11.39% on CIFAR-10 with 4 labels per class. As a point of reference, among the methods studied in [36] (where the same network architecture was used), the lowest error rate achieved on CIFAR-10 with *400* labels per class was 13.13%. Our results also compare favorably to recent state-of-the-art results achieved by ReMixMatch [3], despite the fact that we omit various components such as the self-supervised loss.

Our results are state-of-the-art on all datasets except for CIFAR-100 where ReMixMatch performs a bit better. To understand why ReMixMatch performs better than FixMatch, we experimented with a few variants of FixMatch which copy various components of ReMixMatch into FixMatch. We find that the most important term is Distribution Alignment (DA), which encourages the model predictions to have the same class distribution as the labeled set. Combining FixMatch with DA reaches a **40.14**% error rate with 400 labeled examples, which is substantially better than the 44.28% achieved by ReMixMatch.

We find that in most cases the performance of FixMatch using CTAugment and RandAugment is similar, except in the settings where we have 4 labels per class. This may be explained by the fact that these results are particularly high-variance. For example, the variance over 5 different folds for CIFAR-10 with 4 labels per class is 3.35%, which is significantly higher than that with 25 labels per class (0.33%). The error rates are also affected significantly by the random seeds when the number of labeled examples per class is extremely small, as shown in table 8 of supplementary material.

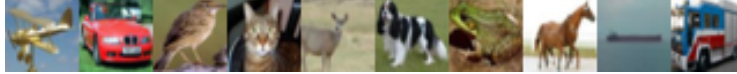

Figure 2: FixMatch reaches 78% CIFAR-10 accuracy using only above 10 labeled images.

## 4.2 STL-10

The STL-10 dataset contains 5,000 labeled images of size 96×96 from 10 classes and 100,000 unlabeled images. There exist out-of-distribution images in the unlabeled set, making it a more realistic and challenging test of SSL performance. We test SSL algorithms on five of the predefined folds of 1,000 labeled images each. Following [4], we use a WRN-37-2 network (comprising 5.9M parameters).[5] As in table 2, FixMatch achieves the state-of-the-art performance of ReMixMatch [3] despite being significantly simpler.

## 4.3 ImageNet

We evaluate FixMatch on ImageNet to verify that it performs well on a larger and more complex dataset. Following [54], we use 10% of the training data as labeled and treat the rest as unlabeled examples. We use a ResNet-50 network architecture and RandAugment [11] as strong augmentation for this experiment. We include additional implementation details in appendix C. FixMatch achieves a top-1 error rate of $28.54 \pm 0.52\%$, which is $2.68\%$ better than UDA [54]. Our top-5 error rate is $10.87 \pm 0.28\%$. While $S^4L$ [57] holds state-of-the-art on semi-supervised ImageNet with a $26.79\%$ error rate, it leverages 2 additional training phases (pseudo-label re-training and supervised fine-tuning) to significantly lower the error rate from $30.27\%$ after the first phase. FixMatch outperforms $S^4L$ after its first phase, and it is possible that a similar performance gain could be achieved by incorporating these techniques into FixMatch.

## 4.4 Barely Supervised Learning

To test the limits of our proposed approach, we applied FixMatch to CIFAR-10 with **only one example per class**.[6] We conduct two sets of experiments.

First, we create four datasets by randomly selecting one example per class. We train on each dataset four times and reach between $48.58\%$ and $85.32\%$ test accuracy with a median of $64.28\%$. The inter-dataset variance is much lower, however; for example, the four models trained on the first dataset all reach between $61\%$ and $67\%$ accuracy, and the second dataset reaches between $68\%$ and $75\%$.

We hypothesize that this variability is caused by the quality of the 10 labeled examples comprising each dataset and that sampling low-quality examples might make it more difficult for the model to learn some particular class effectively. To test this, we construct eight new training datasets with examples ranging in "prototypicality" (i.e., representative of the underlying class). Specifically, we take the ordering of the CIFAR-10 training set from [5] that sorts examples from those that are most representative to those that are least. This example ordering was determined after training many CIFAR-10 models with all labeled data. We thus do not envision this as a practical method for choosing examples for use in SSL, but rather to experimentally verify that examples that are more representative are better suited for low-label training. We divide this ordering evenly into eight buckets (so all of the most representative examples are in the first bucket, and all of the outliers in the last). We then create eight labeled training sets by randomly selecting one labeled example of each class from the same bucket.

Using the same hyperparameters, the model trained only on the most prototypical examples reaches a median of $78\%$ accuracy (with a maximum of $84\%$ accuracy); training on the middle of the distribution reaches $65\%$ accuracy; and training on only the outliers fails to converge completely, with $10\%$ accuracy. Figure 2 shows the full labeled training dataset for the split where FixMatch achieved a median accuracy of $78\%$. Further analysis is presented in Appendix B.7.

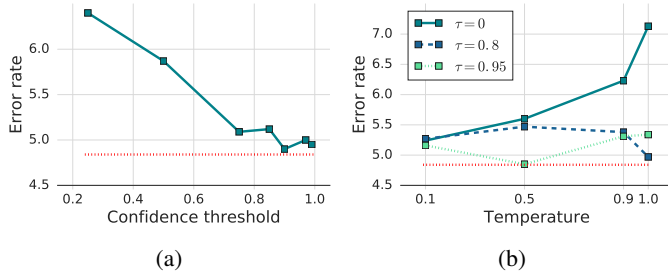

|              | (a) | (b) |
|--------------|-----|-----|

| Ablation | Error |
|----------|-------|
| FixMatch | **4.84** |
| Only Cutout | 6.15 |
| No Cutout | 6.15 |

Figure 3: Plots of ablation studies on FixMatch. (a) Varying the confidence threshold for pseudo-labels. (b) Measuring the effect of "sharpening" the predicted label distribution while varying the confidence threshold ($\tau$). Error rate of FixMatch with default hyperparameters is in red dotted line.

Table 3: Ablation study with different strong data augmentation of FixMatch. Error rates are reported on a single 250-label split from CIFAR-10.

## 5 Ablation Study

Since FixMatch comprises a simple combination of two existing techniques, we perform an extensive ablation study to better understand why it is able to obtain state-of-the-art results. Due to the number of experiments in our ablation study, we focus on studying with a single 250 label split from CIFAR-10 and only report results using CTAugment. Note that FixMatch with default parameters achieves $4.84\%$ error rate on this particular split. We present complete ablation results, including optimizer (appendix B.3), learning rate decay schedule (appendix B.4), weight decay (appendix B.6), labeled to unlabeled data ratio $\mu$ (appendix B.5), in the supplementary material.

### 5.1 Sharpening and Thresholding

A "soft" version of pseudo-labeling can be designed by sharpening the predicted distribution. This formulation appears in UDA and is of general interest since other methods such as MixMatch and ReMixMatch also make use of sharpening (albeit without thresholding). Using sharpening instead of an $\arg\max$ introduces a hyper-parameter: the temperature $T$ [4, 54, 3].

We study the interactions between the temperature $T$ and the confidence threshold $\tau$. Note that pseudo-labeling in FixMatch is recovered as $T \to 0$. The results are presented in fig. 3a and fig. 3b. The threshold value of $0.95$ shows the lowest error rate, though increasing it to $0.97$ or $0.99$ did not hurt much. In contrast, accuracy drops by more than 1.5% when using a small threshold value. Note that the threshold value controls the trade-off between the quality and the quantity of pseudo-labels. As discussed in appendix B.2, the accuracy of pseudo-labels for unlabeled data increases with higher threshold values, while the amount of unlabeled data contributing to $\ell_u$ in eq. (4) decreases. This suggests that the quality of pseudo-labels is more important than the quantity for reaching a high accuracy. Sharpening, on the other hand, did not show a significant difference in performance when a confidence threshold is used. In summary, we observe that swapping pseudo-labeling for sharpening and thresholding would introduce a new hyperparameter while achieving no better performance.

### 5.2 Augmentation Strategy

We conduct an ablation study on different strong data augmentation policies as it plays a key role in FixMatch. Specifically, we chose RandAugment [11] and CTAugment [3], which have been used for state-of-the-art SSL algorithms such as UDA [54] and ReMixMatch [4] respectively. On CIFAR-10, CIFAR-100, and SVHN we observed highly comparable results between the two policies, whereas in STL-10 (table 2), we observe a significant gain by using CTAugment.

We measure the effect of Cutout in table 3, which is used by default after strong augmentation in both RandAugment and CTAugment. We find that both Cutout and CTAugment are required to obtain the best performance; removing either results in a significant increase in error rate.

We also study different combinations of weak and strong augmentations for pseudo-label generation and prediction (i.e., the upper and lower paths in fig. 1). When we replaced the weak augmentation for label guessing with strong augmentation, we found that the model diverged early in training. Conversely, when replacing weak augmentation with *no* augmentation, the model overfits the guessed unlabeled labels. Using weak augmentation in place of strong augmentation to generate the model's prediction for training peaked at 45% accuracy but was not stable and progressively collapsed to 12%,

suggesting the importance of strong data augmentation. This observation is well-aligned with those from supervised learning [10].

## 6 Conclusion

There has been rapid recent progress in SSL. Unfortunately, much of this progress comes at the cost of increasingly complicated learning algorithms with sophisticated loss terms and numerous difficult-to-tune hyper-parameters. We introduce FixMatch, a simpler SSL algorithm that achieves state-of-the-art results across many datasets. We show how FixMatch can begin to bridge the gap between low-label semi-supervised learning and few-shot learning or clustering: we obtain surprisingly-high accuracy with just one label per class. Using only standard cross-entropy losses on both labeled and unlabeled data, FixMatch's training objective can be written in just a few lines of code.

Because of this simplicity, we are able to thoroughly investigate how FixMatch works. We find that certain design choices are important (and often underemphasized) – most importantly, weight decay and the choice of optimizer. The importance of these factors means that even when controlling for model architecture as is recommended in [36], the same technique can not always be directly compared across different implementations.

On the whole, we believe that the existence of such simple but performant semi-supervised machine learning algorithms will help to allow machine learning to be deployed in increasingly many practical domains where labels are expensive or difficult to obtain.

## Broader Impact

FixMatch helps democratize machine learning in two ways: first, its simplicity makes it available to a wider audience, and second, its accuracy with only a few labels means that it can be applied to domains where previously machine learning was not feasible. The flip side of democratization of machine learning research is that it becomes easy for both good and bad actors to apply. We hope that this ability will be used for good—for example, obtaining medical scans is often far cheaper than paying an expert doctor to label every image. However, it is possible that more advanced techniques for semi-supervised learning will allow for more advanced surveillance: for example, the efficacy of our one-shot classification might allow for more accurate person identification from a few images. Broadly speaking, any progress on semi-supervised learning will have these same consequences.

## Funding Disclosure

Google is the sole source of funding for this work.

## Acknowledgment

We thank Qizhe Xie, Avital Oliver, Quoc V. Le, and Sercan Arik for their feedback on this paper.

## Footnotes

[2]In practice, we include all labeled data as part of unlabeled data without their labels when constructing $\mathcal{U}$.

[3]https://www.pythonware.com/products/pil/

[4]$\beta$ refers to a momentum in SGD optimizer. The definition of other hyperparameters are found in section 2.

[5]We clarify that both FixMatch and ReMixMatch [3], which has reported an incorrect number of network parameters (23.8M), are tested with the same network architecture containing 5.9M parameters.

[6]The experimental protocol of barely supervised learning (BSL) shares similarities to those of few-shot learning (FSL) [37] as they both assume a limited availability (e.g., 1 or 5) of labeled examples from categories of interest. However, two protocols have a critical difference, where for FSL one is provided with extra labeled training examples from regular classes, whereas for BSL one is given additional unlabeled training examples.

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
