[Supplementary Material]

# A   Algorithm

We present the complete algorithm for FixMatch in algorithm 1.

---

**Algorithm 1** FixMatch algorithm.

---

1: **Input:** Labeled batch $\mathcal{X} = \big\{(x_b, p_b) : b \in (1, \ldots, B)\big\}$, unlabeled batch $\mathcal{U} = \big\{u_b : b \in (1, \ldots, \mu B)\big\}$, confidence threshold $\tau$, unlabeled data ratio $\mu$, unlabeled loss weight $\lambda_u$.
2: $\ell_s = \frac{1}{B} \sum_{b=1}^{B} \mathrm{H}(p_b, \alpha(x_b))$ {*Cross-entropy loss for labeled data*}
3: **for** $b = 1$ **to** $\mu B$ **do**
4:     $q_b = p_{\mathrm{m}}(y \mid \alpha(u_b); \theta)$ {*Compute prediction after applying weak data augmentation of $u_b$*}
5: **end for**
6: $\ell_u = \frac{1}{\mu B} \sum_{b=1}^{\mu B} \mathbb{1}\{\max(q_b) > \tau\} \, \mathrm{H}(\arg\max(q_b), p_{\mathrm{m}}(y \mid \mathcal{A}(u_b))$ {*Cross-entropy loss with pseudo-label and confidence for unlabeled data*}
7: **return** $\ell_s + \lambda_u \ell_u$

---

# B   Comprehensive Experimental Results

## B.1   Hyperparameters

As mentioned in section 4, we used almost identical hyperparameters of FixMatch on CIFAR-10, CIFAR-100, SVHN and STL-10. Note that we used similar network architectures for these datasets, except that more convolution filters were used for CIFAR-100 (WRN-28-8) to handle larger label space and more convolutions were used for STL-10 (WRN-37-2) to deal with larger input image size. Following the suggestion in [4], we doubled the weight decay parameter for WRN-28-8 to avoid overfitting. Here, we provide a complete list of hyperparameters in table 4. Note that we did ablation study for most of these hyperparameters in section 5 ($\tau$ in section 5.1, $\mu$ in appendix B.5, $lr$ and $\beta$ (momentum) in appendix B.3, and weight decay in appendix B.6).

| | CIFAR-10 | CIFAR-100 | SVHN | STL-10 |
|---|---|---|---|---|
| $\tau$ | | 0.95 | | |
| $\lambda_u$ | | 1 | | |
| $\mu$ | | 7 | | |
| $B$ | | 64 | | |
| $lr$ | | 0.03 | | |
| $\beta$ | | 0.9 | | |
| Nesterov | | True | | |
| weight decay | 0.0005 | 0.001 | 0.0005 | 0.0005 |

Table 4: Complete list of FixMatch hyperparameters for CIFAR-10, CIFAR-100, SVHN and STL-10.

## B.2   Trade-off between the Quality and the Quantity of Pseudo-Labels with Confidence

To better understand the role of thresholding in FixMatch, we present in table 5 two additional measurements along with the test set accuracy: the impurity (the error rate of unlabeled data that falls above the threshold) and the mask rate (the number of examples which are masked out) which are computed as follows:

$$\text{impurity} = \frac{\sum_{b=1}^{\mu B} \mathbb{1}(\max(q_b) \geq \tau)\mathbb{1}(y_b \neq \hat{q}_b)}{\sum_{b=1}^{\mu B} \mathbb{1}(\max(q_b) \geq \tau)} \tag{5}$$

$$\text{mask rate} = \frac{1}{\mu B} \sum_{b=1}^{\mu B} \mathbb{1}(\max(q_b) \geq \tau) \tag{6}$$

As shown in table 5, when using small threshold values, most unlabeled examples' confidence is above the threshold. Consequently, they all contribute to the unlabeled loss in eq. (4). Unfortunately, pseudo-labels of these examples are not always correct and the learning process is significantly

| $\tau$ | mask rate | impurity | error rate |
|---|---|---|---|
| 0.25 | 100.00 | 6.39 | 6.40 |
| 0.5 | 100.00 | 5.40 | 5.87 |
| 0.75 | 99.82 | 5.35 | 5.09 |
| 0.85 | 99.31 | 4.32 | 5.12 |
| 0.9 | 99.21 | 3.85 | 4.90 |
| 0.95 | 98.13 | 3.47 | 4.84 |
| 0.97 | 96.35 | 2.30 | 5.00 |
| 0.99 | 92.14 | 2.06 | 5.05 |

Table 5: The mask rate and impurity at the end of the training along with the test set error rate of FixMatch using different threshold values on a single 250-label split from CIFAR-10.

| Decay Schedule | Error |
|---|---|
| Cosine (FixMatch) | 4.84 |
| Linear Decay (end 0.01) | 4.95 |
| Linear Decay (end 0.02) | 5.55 |
| No Decay | 5.70 |

Table 6: Ablation study on learning rate decay schedules. Error rates are reported on a single 250-label split from CIFAR-10.

impeded by noisy pseudo-labeled examples. This behavior is known as confirmation bias [1]. On the other hand, using high threshold values allows a smaller fraction of ostensibly higher-quality unlabeled examples to contribute to the unlabeled loss, effectively reducing the confirmation bias with strong data augmentation, resulting in lower error rates on the test set. Given our observation on the trade-off between the quality and the quantity of pseudo-labels, combining improved techniques for confidence calibration and uncertainty estimation [18, 27, 26, 19] into FixMatch would be a promising future direction.

## B.3 Ablation Study on Optimizer

While the study of different optimizers and their hyperparameters is seldom done in previous SSL works, we found that they can have a strong effect on performance. We present ablation results on optimizers in table 7. First, we studied the effect of momentum ($\beta$) for the SGD optimizer. We found that the performance is somewhat sensitive to $\beta$ and the model did not converge when $\beta$ is set too large. On the other hand, small values of $\beta$ still worked fine. When $\beta$ is small, increasing the learning rate improved the performance, though they are not as good as the best performance obtained with $\beta = 0.9$. Nesterov momentum resulted in a slightly lower error rate than that of standard momentum SGD, but the difference was not significant.

As studied in [53, 29], we did not find Adam performing better than momentum SGD. While the best error rate of the model trained with Adam is only 0.53% larger than that of momentum SGD, we found that the performance was much more sensitive to the change of learning rate (e.g., increase in error rate by more than 8% when increasing the learning rate to 0.002) than momentum SGD. Additional exploration along this direction to make Adam more competitive includes the use of weight decay [29, 58] instead of L2 weight regularization and a better exploration of hyperparameters [7, 8].

## B.4 Ablation Study on Learning Rate Schedule

It is a popular choice in recent works [28] to use a cosine learning rate decay. As shown in table 6, a linear learning rate decay performed nearly as well. Note that, as for the cosine learning rate decay, picking a proper decaying rate is important. Finally, using no decay results in worse accuracy (a 0.86% degradation).

| Optimizer | Hyperparameters | | | Error |
|---|---|---|---|---|
| SGD | $\eta = 0.03$ | $\beta = 0.90$ | Nesterov | **4.84** |
| SGD | $\eta = 0.03$ | $\beta = 0.999$ | Nesterov | 84.33 |
| SGD | $\eta = 0.03$ | $\beta = 0.99$ | Nesterov | 21.97 |
| SGD | $\eta = 0.03$ | $\beta = 0.50$ | Nesterov | 5.79 |
| SGD | $\eta = 0.03$ | $\beta = 0.25$ | Nesterov | 6.42 |
| SGD | $\eta = 0.03$ | $\beta = 0$ | Nesterov | 6.76 |
| SGD | $\eta = 0.05$ | $\beta = 0$ | Nesterov | 6.06 |
| SGD | $\eta = 0.10$ | $\beta = 0$ | Nesterov | 5.27 |
| SGD | $\eta = 0.20$ | $\beta = 0$ | Nesterov | 5.19 |
| SGD | $\eta = 0.50$ | $\beta = 0$ | Nesterov | 5.74 |
| SGD | $\eta = 0.03$ | $\beta = 0.90$ | | **4.86** |
| Adam | $\eta = 0.002$ | $\beta_1 = 0.9$ | $\beta_2 = 0.00$ | 29.42 |
| Adam | $\eta = 0.002$ | $\beta_1 = 0.9$ | $\beta_2 = 0.90$ | 14.42 |
| Adam | $\eta = 0.002$ | $\beta_1 = 0.9$ | $\beta_2 = 0.99$ | 15.44 |
| Adam | $\eta = 0.002$ | $\beta_1 = 0.9$ | $\beta_2 = 0.999$ | 13.93 |
| Adam | $\eta = 0.0008$ | $\beta_1 = 0.9$ | $\beta_2 = 0.999$ | 7.35 |
| Adam | $\eta = 0.0006$ | $\beta_1 = 0.9$ | $\beta_2 = 0.999$ | 6.12 |
| Adam | $\eta = 0.0005$ | $\beta_1 = 0.9$ | $\beta_2 = 0.999$ | 5.95 |
| Adam | $\eta = 0.0004$ | $\beta_1 = 0.9$ | $\beta_2 = 0.999$ | 5.44 |
| Adam | $\eta = 0.0003$ | $\beta_1 = 0.9$ | $\beta_2 = 0.999$ | 5.37 |
| Adam | $\eta = 0.0002$ | $\beta_1 = 0.9$ | $\beta_2 = 0.999$ | 5.57 |
| Adam | $\eta = 0.0001$ | $\beta_1 = 0.9$ | $\beta_2 = 0.999$ | 7.90 |

Table 7: Ablation study on optimizers. Error rates are reported on a single 250-label split from CIFAR-10.

Figure 4: Plots of ablation studies on optimizers. (a) Varying $\beta$. (b) Varying $\eta$ with $\beta = 0$.

## B.5 Ratio of Labeled to Unlabeled Data in Minibatch

In fig. 5a we plot the error rates of FixMatch with different ratios of unlabeled data ($\mu$) in each minibatch. We observe a significant decrease in error rates by using a large amount of unlabeled data, which is consistent with the finding in UDA [54]. In addition, scaling the learning rate $\eta$ linearly with the batch size (a technique for large-batch supervised training [16]) was effective for FixMatch, especially when $\mu$ is small.

## B.6 Weight Decay

While the value 0.0005 appeared as a good default choice for WRN-28-2 across datasets, we find that the weight decay could have a huge impact on performance when tuned incorrectly for low label regimes: choosing a value that is just one order of magnitude larger or smaller than optimal can cost ten percentage points or more, as shown in fig. 5b.

## B.7 Labeled Data for Barely Supervised Learning

In addition to fig. 2, we visualize the full labeled training images obtained by ordering mechanism [5] used for barely supervised learning in fig. 6. Each row contains 10 images from 10 different classes

Figure 5: Plots of ablation studies on FixMatch. (a) Varying the ratio of unlabeled data ($\mu$) with different learning rate ($\eta$) scaling strategies. (b) Varying the loss coefficient for weight decay. Error rate of FixMatch with default hyperparameters is in red dotted line.

Figure 6: Labeled training data for the 1-label-per-class experiment. Each row corresponds to the complete labeled training set for one run of our algorithm, sorted from the most prototypical dataset (first row) to least prototypical dataset (last row).

of CIFAR-10 and is used as the complete labeled training dataset for one run of FixMatch. The first row contains the most prototypical images of each class, while the bottom row contains the least prototypical images. We train two models for each dataset and compute the mean accuracy between the two and plot this in fig. 7. Observe that we obtain over 80% accuracy when training on the best examples.

### B.8 Comparison to Supervised Baselines

In table 9 and table 10, we present the performance of models trained only with the labeled data using strong data augmentations to highlight the effectiveness of using unlabeled data in FixMatch.

## C  Implementation Details for Section 4.3

For our ImageNet experiments we use standard ResNet50 pre-activation model trained in a distributed way on a TPU device with 32 cores.[7] We report results over five random folds of labeled data. We use the following set of hyperparameters for our ImageNet model:

- **Batch size**. On each step our batch contains 1024 labeled examples and 5120 unlabeled examples.

| Dataset | 1 | 2 | 3 | 4 | 5 |
|---|---|---|---|---|---|
| CIFAR-10 | 5.46 | 6.17 | 9.37 | 10.85 | 13.32 |
| SVHN | 2.40 | 2.47 | 6.24 | 6.32 | 6.38 |

Table 8: Error rates of FixMatch (CTA) on a single 40-label split of CIFAR-10 and SVHN with different random seeds. Runs are ordered by accuracy.

| | CIFAR-10 | | | CIFAR-100 | | | SVHN | | |
|---|---|---|---|---|---|---|---|---|---|
| Method | 40 labels | 250 labels | 4000 labels | 400 labels | 2500 labels | 10000 labels | 40 labels | 250 labels | 1000 labels |
| Supervised (RA) | 64.01±0.76 | 39.12±0.77 | 12.74±0.29 | 79.47±0.18 | 52.88±0.51 | 32.55±0.21 | 52.68±2.29 | 22.48±0.55 | 10.89±0.12 |
| Supervised (CTA) | 64.53±0.83 | 41.92±1.17 | 13.64±0.12 | 79.79±0.59 | 54.23±0.48 | 35.30±0.19 | 43.05±2.34 | 15.06±1.02 | 7.69±0.27 |
| FixMatch (RA) | 13.81±3.37 | 5.07±0.65 | 4.26±0.05 | 48.85±1.75 | 28.29±0.11 | 22.60±0.12 | 3.96±2.17 | 2.48±0.38 | 2.28±0.11 |
| FixMatch (CTA) | 11.39±3.35 | 5.07±0.33 | 4.31±0.15 | 49.95±3.01 | 28.64±0.24 | 23.18±0.11 | 7.65±7.65 | 2.64±0.64 | 2.36±0.19 |

Table 9: Error rates for CIFAR-10, CIFAR-100 and SVHN on 5 different folds. Models with (RA) uses RandAugment [11] and the ones with (CTA) uses CTAugment [3] for strong-augmentation. All models are tested using the same codebase.

Figure 7: Accuracy of the model when trained on the 1-label-per-class datasets from Figure 6, ordered from most prototypical (top row) to least (bottom row).

| Method | Error rate | Method | Error rate |
|---|---|---|---|
| Supervised (RA) | 20.66±0.83 | FixMatch (RA) | 7.98±1.50 |
| Supervised (CTA) | 19.86±0.66 | FixMatch (CTA) | 5.17±0.63 |

Table 10: Error rates for STL-10 on 1000-label splits. All models are tested using the same codebase.

- **Training time**. We train our model for 300 epochs of unlabeled examples[8].
- **Learning rate schedule**. We utilize linear learning rate warmup for the first 5 epochs until it reaches an initial value of $0.4$. Then we the decay learning rate at epochs 60, 120, 160, and 200 epoch by multiplying it by $0.1$.
- **Optimizer**. We use Nesterov Momentum optimizer with momentum $0.9$.
- **Exponential moving average (EMA)**. We utilize EMA technique with decay $0.999$.
- **FixMatch loss**. We use unlabeled loss weight $\lambda_u = 10$ and confidence threshold $\tau = 0.7$ in FixMatch loss.
- **Weight decay**. Our weight decay coefficient is $0.0003$. Similarly to other datasets we perform weight decay by adding L2 penalty of all weights to model loss.
- **Augmentation of unlabeled images**. For strong augmentation we use RandAugment with random magnitude [11]. For weak augmentation we use a random horizontal flip.
- **ImageNet preprocessing**. We randomly crop and rescale to $224\times224$ size all labeled and unlabeled training images prior to performing augmentation. This is considered a standard ImageNet preprocessing technique.

# D   Extensions of FixMatch

## D.1   Augmentation Anchoring and Distribution Alignment

Augmentation Anchoring, where $M$ strong augmentations are applied to each unlabeled example for consistency regularization, and Distribution Alignment, which encourages the model predictions to have same class distribution as the labeled set, are two important techniques to the success of ReMixMatch [3]. Thanks to its simplicity and clean formulation, incorporating these techniques into FixMatch becomes straightforward. Firstly, we incorporate Augmentation Anchoring into FixMatch as follows:

$$\ell_u = \frac{1}{\mu B} \sum_{b=1}^{\mu B} \mathbb{1}(\max(q_b) \geq \tau) \times \frac{1}{M} \sum_{i=1}^{M} \mathrm{H}(\hat{q}_b, p_\mathrm{m}(y \mid \mathcal{A}(u_b))) \tag{7}$$

Note that the strong augmentation $\mathcal{A}(u_b)$ is a stochastic process and it produces $M$ different strongly-augmented examples of an unlabeled example $u_b$. Using $M = 4$ and $\mu = 4$, FixMatch (CTA) with Augmentation Anchoring reduces the error rate averaged over 5 different folds on CIFAR-10 with 250 labeled examples from $5.07\%$ to $4.81\%$.

As already reported in section 4.1, combining Distribution Alignment into FixMatch improves the SSL performance significantly, especially when the number of labeled training data is very limited. Specifically, we align the predictive distribution of a weakly-augmented example $q_b = p_m(y|\alpha(u_b))$ using the dataset's marginal class distribution estimated using the labeled data and the running average of the model's prediction estimated by the unlabeled data as follows:

$$\tilde{q}_b = \mathrm{Normalize}\left( q_b \times \frac{p(y|\mathcal{X})}{p_m(y|\mathcal{U})} \right) \tag{8}$$

where $\mathrm{Normalize}(x)_i = x_i / \sum_j x_j$. Now, eq. (4) can be modified as follows accordingly:

$$\ell_u = \frac{1}{\mu B} \sum_{b=1}^{\mu B} \mathbb{1}(\max(\tilde{q}_b) \geq \tau) \,\mathrm{H}(\hat{\tilde{q}}_b, p_\mathrm{m}(y \mid \mathcal{A}(u_b))) \tag{9}$$

FixMatch (CTA) with distribution alignment reduces the error rate averaged over 5 different folds on CIFAR-10 with 40 labeled examples from $11.38\%$ to $9.47\%$. On CIFAR-100 with 400 labeled examples, it reduces the error rate from $49.95\%$ to $40.14\%$, which is also lower than $44.28\%$ of ReMixMatch. In addition to Section 4.3, we conduct SSL experiments on ImageNet using 1% of its data as labeled examples. We find that in this regime the role of distribution alignment becomes more critical – FixMatch model does not train well without distribution alignment. On the other hand, after a proper tuning of hyperparameters (weight of unlabeled loss $\lambda_u = 3$ and confidence threshold $\tau = 0.9$), FixMatch (RA) model with distribution alignment achieves $67.1\%$ top-1 and $47.7\%$ top-5 error rate (this correspond to $32.9\%$ top-1 and $52.3\%$ top-5 accuracy and similar to [57] results).

## D.2   Datatype-Agnostic Data Augmentation

Strong augmentation plays a key role in FixMatch. Applying FixMatch to different problem domains than vision thus requires one to come up with a novel augmentation strategy. While there are domain-specific data augmentation strategies for different application domains, such as back-translation [48] for text classification or SpecAugment [38] for speech recognition, it is desirable if FixMatch can also be combined with datatype-agnostic data augmentation methods.

In this section, we consider two such augmentation schemes, namely MixUp [59] and Virtual Adversarial Training (VAT) [33], as a replacement of RandAugment or CTAugment in FixMatch for image classification. For MixUp, instead of mixing both input and label, we mixed random pairs of inputs only using $\alpha = 9$ to be more consistent with the data augmentation in FixMatch. For VAT, we used $\tau = 0.5$. We evaluated on CIFAR-10 with 250 labeled data protocol and report the mean and standard deviation over 5 different folds in table 11. We make comparisons of our FixMatch variants to MixMatch [4] and VAT [33]. The FixMatch variant with (input) MixUp obtained comparable error rates to MixMatch, while the variant with VAT achieved significantly lower error rates than VAT. This suggests the generality of the FixMatch's framework against different data augmentation strategies.

| FixMatch + Input MixUp | MixMatch | FixMatch + VAT | VAT |
|---|---|---|---|
| 10.99±0.50 | 11.05±0.86 | 32.26±2.24 | 36.03±2.82 |

Table 11: Error rates for CIFAR-10 with 250 labeled examples on 5 different folds. All models are tested using the same codebase.

# E  List of Data Transformations

Given a collection of transformations (e.g., color inversion, translation, contrast adjustment, etc.), RandAugment randomly selects transformations for each sample in a mini-batch. As originally proposed, RandAugment uses a single fixed global magnitude that controls the severity of all distortions [11]. The magnitude is a hyperparameter that must be optimized on a validation set e.g., using grid search. We found that sampling a random magnitude from a pre-defined range at each training step (instead of using a fixed global value) works better for semi-supervised training, similar to what is used in UDA [54].

Instead of setting the transformation magnitudes randomly, CTAugment [3] learns them online over the course of training. To do so, a wide range of transformation magnitude values is divided into bins (as in AutoAugment [10]) and a weight (initially set to 1) is assigned to each bin. All examples are augmented with a pipeline consisting of two transformations which are sampled uniformly at random. For a given transformation, a magnitude bin is sampled randomly with a probability according to the (normalized) bin weights. To update the weights of the magnitude bins, a labeled example is augmented with two transformations whose magnitude bins are sampled uniformly at random. The magnitude bin weights are then updated according to how close the model's prediction is to the true label. Further details on CTAugment can be found in [3].

We used the same sets of image transformations used in RandAugment [11] and CTAugment [3]. For completeness, we listed all transformation operations for these augmentation strategies in table 12 and table 13.

| Transformation | Description | Parameter | Range |
|---|---|---|---|
| Autocontrast | Maximizes the image contrast by setting the darkest (lightest) pixel to black (white). | | |
| Brightness | Adjusts the brightness of the image. $B = 0$ returns a black image, $B = 1$ returns the original image. | $B$ | [0.05, 0.95] |
| Color | Adjusts the color balance of the image like in a TV. $C = 0$ returns a black & white image, $C = 1$ returns the original image. | $C$ | [0.05, 0.95] |
| Contrast | Controls the contrast of the image. A $C = 0$ returns a gray image, $C = 1$ returns the original image. | $C$ | [0.05, 0.95] |
| Equalize | Equalizes the image histogram. | | |
| Identity | Returns the original image. | | |
| Posterize | Reduces each pixel to $B$ bits. | $B$ | [4, 8] |
| Rotate | Rotates the image by $\theta$ degrees. | $\theta$ | [-30, 30] |
| Sharpness | Adjusts the sharpness of the image, where $S = 0$ returns a blurred image, and $S = 1$ returns the original image. | $S$ | [0.05, 0.95] |
| Shear_x | Shears the image along the horizontal axis with rate $R$. | $R$ | [-0.3, 0.3] |
| Shear_y | Shears the image along the vertical axis with rate $R$. | $R$ | [-0.3, 0.3] |
| Solarize | Inverts all pixels above a threshold value of $T$. | $T$ | [0, 1] |
| Translate_x | Translates the image horizontally by ($\lambda \times$ image width) pixels. | $\lambda$ | [-0.3, 0.3] |
| Translate_y | Translates the image vertically by ($\lambda \times$ image height) pixels. | $\lambda$ | [-0.3, 0.3] |

Table 12: List of transformations used in RandAugment [11].

| Transformation | Description | Parameter | Range |
|---|---|---|---|
| Autocontrast | Maximizes the image contrast by setting the darkest (lightest) pixel to black (white), and then blends with the original image with blending ratio $\lambda$. | $\lambda$ | [0, 1] |
| Brightness | Adjusts the brightness of the image. $B = 0$ returns a black image, $B = 1$ returns the original image. | $B$ | [0, 1] |
| Color | Adjusts the color balance of the image like in a TV. $C = 0$ returns a black & white image, $C = 1$ returns the original image. | $C$ | [0, 1] |
| Contrast | Controls the contrast of the image. A $C = 0$ returns a gray image, $C = 1$ returns the original image. | $C$ | [0, 1] |
| Cutout | Sets a random square patch of side-length ($L \times$image width) pixels to gray. | $L$ | [0, 0.5] |
| Equalize | Equalizes the image histogram, and then blends with the original image with blending ratio $\lambda$. | $\lambda$ | [0, 1] |
| Invert | Inverts the pixels of the image, and then blends with the original image with blending ratio $\lambda$. | $\lambda$ | [0, 1] |
| Identity | Returns the original image. | | |
| Posterize | Reduces each pixel to $B$ bits. | $B$ | [1, 8] |
| Rescale | Takes a center crop that is of side-length ($L \times$image width), and rescales to the original image size using method $M$. | $L$ | [0.5, 1.0] |
| | | $M$ | see caption |
| Rotate | Rotates the image by $\theta$ degrees. | $\theta$ | [-45, 45] |
| Sharpness | Adjusts the sharpness of the image, where $S = 0$ returns a blurred image, and $S = 1$ returns the original image. | $S$ | [0, 1] |
| Shear_x | Shears the image along the horizontal axis with rate $R$. | $R$ | [-0.3, 0.3] |
| Shear_y | Shears the image along the vertical axis with rate $R$. | $R$ | [-0.3, 0.3] |
| Smooth | Adjusts the smoothness of the image, where $S = 0$ returns a maximally smooth image, and $S = 1$ returns the original image. | $S$ | [0, 1] |
| Solarize | Inverts all pixels above a threshold value of $T$. | $T$ | [0, 1] |
| Translate_x | Translates the image horizontally by ($\lambda \times$image width) pixels. | $\lambda$ | [-0.3, 0.3] |
| Translate_y | Translates the image vertically by ($\lambda \times$image height) pixels. | $\lambda$ | [-0.3, 0.3] |

Table 13: List of transformations used in CTAugment [3]. The ranges for the listed parameters are discretized into 17 equal bins, except for the $M$ parameter of the Rescale transformation, which takes one of the following six options: anti-alias, bicubic, bilinear, box, hamming, and nearest.

## Footnotes

[7] https://github.com/tensorflow/tpu/tree/master/models/official/resnet

[8]Note that one epoch of unlabelled examples contains all 1.2 million examples from Imagenet training set and it correspond to 10 passes through labelled set for 10% Imagenet task.