[Reviews · NeurIPS 2020]

Review 1

Summary and Contributions: The paper proposes a rather simple but efficient algorithm for semi-supervised learning. The algorithm is based on the previously proposed teacher-student architecture. The novelty is 1) that the teacher always receives weakly augmented samples (flip and shift) and the student receives strongly augmented samples (RandAugment and CTAugment proposed previously); 2) instead of computing the loss for all unlabeled examples, the loss is computed only for unlabeled examples for which the teacher is confident, the target for the student is one-hot coded label instead of the distribution (as was done previously).

Strengths: - The proposed algorithm is simple and efficient. The experimental results are good. - It is very nice that one does not have to use ramp-ups for the weight of the consistency cost. The proposed solution is more elegant. - I liked the ablation study on optimisers (Table 7 in the appendix). That study shows quite high sensitivity of the SSL performance on the parameters of the optimizer.

Weaknesses: - The results are similar to the previous state of the art. The proposed method seems like a small modification of the existing algorithms. Below are some points that apply to many SSL papers published recently: - It is unclear whether the proposed algorithm can be extended to other domains (not image classification). - I wonder how much of the success of this and other similar SSL methods is due to our knowledge of the domain (image classification) that comes from training on large labeled data sets. Specifically we know what architectural choices work and do not work on particular datasets when training with many labels. One indicator of that is the usage of different hyperparameters for the smaller datasets and ImageNet in the paper. - Is the scenario considered in the paper realistic for many practical applications? I think that in most applications, the unlabeled samples do not come from the same classes.

Correctness: The methodology is consistent with the recently published SSL papers. However, the ablation study on optimisers suggests that the choice of the optimiser has large impact on the results. In the light of this result, it seems that this paper (and most other SSL papers) overfit to the (large labeled) test set. Is it so?

Clarity: The paper is well written.

Relation to Prior Work: Yes.

Reproducibility: Yes

Additional Feedback: - Equation (1) is difficult to understand because the two terms look exactly the same. - I would like to see more details on the loss function used in the experiment reported in Fig. 3b.


Review 2

Summary and Contributions: The work combines the pseudo-labeling and the consistency regularization techniques in SSL to propose a simple method for high-performing semi-supervised learning. The main idea is to use weakly-augmented images to produce pseudo-labels, and then train the model on heavy-augmented images with these labels.

Strengths: The paper presents a simplified version of earlier works, such as UDA and ReMixMatch, while achieving performance on par with ReMixMatch. Additionally, some experiments in applying the method to few-shot learning task are presented. An extensive ablation study (in supplementary material) is provided.

Weaknesses: 1. I cite from ReMixMatch figure caption: "Augmentation anchoring. We use the prediction for a weakly augmented image (green, middle) as the target for predictions on strong augmentations of the same image". This sounds to me as a summary of the presented work, and as such I consider it a special case of the ReMixMatch. Authors have discussed the differences between their work and ReMixMatch, mentioning that (1) "ReMixMatch don`t use pseudo labeling", and (2) ReMixMatch uses sharpening of pseudolabels and weight annealing of the unlabeled data loss. However, in section 3.2.1 of ReMixMatch, it is stated that the guessed labels are used as targets (for strongly augmented images) using cross-entropy loss. I believe this is called self-training with pseudo-labeling, just as this work proposes. 2. It is stated (lines 213-215) that FixMatch substantially outperforms MixMatch, ReMixMatch, and UDA with 40 and 250 labels, but this is incorrect. The performance of ReMixMatch is very close to the FixMAtch in this regime (and outperforms the FixMatch with more data). 3. The "Barely supervised learning" section describes 1-shot experiments in some setting that approximates the standard few-shot training/test regime (i.e., with episodes). The authors are encouraged to align with standard few-shot protocols and compare their performance to other methods in the data-starved regime (e.g., ReMixMatch).

Correctness: The correctness of the presented expressions seems to be fine.

Clarity: The paper is clearly written with detailed explanations.

Relation to Prior Work: The discussion of prior art is rich, as it is an important part of this paper.

Reproducibility: Yes

Additional Feedback: The presented work has clear practical value, as it distills the simple and powerful techniques in SSL that deliver SOTA results. However, I find the novelty limited due to the fact these techniques were presented in earlier frameworks, and it is not clear how are they more complicated and difficult to manage.


Review 3

Summary and Contributions: They propose a new approach for semi-supervised learning (SSL) that gives the pseudo-label to a strongly-augmented image using the pseudo-label of weakly-augmented image. Despite the simplicity of their method, they achieve very strong results in SSL benchmark settings.

Strengths: 1. Their proposed method can be a good direction for semi-supervised learning. Although there are several SSL methods effectively using data augmentation such as mix-up, the proposed new approach seems to have different aspects from previous works. The FixMatch is a simple, yet effective SSL method. 2. Empirical evaluation is very carefully designed. The evaluation sufficiently shows the effectiveness of their approach. 3. Analysis on augmentation strategy and sharpening also provide good insights.

Weaknesses: 1. The method does not have good explanation on why guiding predictions for strongly-augmented images by weakly-augmented images works so well. Although this issue can be common with other works, it is good if they provide empirical or theoretically analysis on this point. 2. Is it always easy to define "weak" and "strong" augmentation? They defined two kinds of augmentation in a heuristic way. But, maybe for some datasets, their defined "weak" augmentation can be a "strong" augmentation? I cannot come up with a good example, but the way of data augmentation can be different from datasets to datasets.

Correctness: They are correct.

Clarity: The paper is very well written. It is very easy to follow.

Relation to Prior Work: They provide clear distinction from other works including pre-print works. Their contribution is clearly stated.

Reproducibility: Yes

Additional Feedback:


Review 4

Summary and Contributions: This paper proposes a method called FixMatch for SSL. It simply treats predictions of weak augmented images as pseudo labels with argmax operation, and manages to minimize the loss between pseudo-label and predictions from strong augmented images. The method achieves promising results on several image recognition datasets.

Strengths: 1. The paper is well written and easy to follow. 2. The performances on several datasets are promising

Weaknesses: 1. The novelty of the proposed method is incremental. Both hard pseudo labels and consistency ideas are proposed in previous literatures. 2. Why does hard pseudo-labeling perform better than soft sharpening ?  The soft-labels are widely used in image classification tasks (in Both supervised and SSL settings) to improve the model generalization capacity. When there are only a few labels, there is a high chance that predictions of unlabeled samples are incorrect. The wrong predictions may lead to more noise labels compared to soft labels. 3. The performance on CIFAR100 of the proposed method is inferior to ReMixMatch. The authors demonstrate that the model can achieve best performance with distribution alignment. Why does the result vary so heavily? Is this happening on ImageNet too? Is this related to the number of classes?   Minor Comments 1. It would be better if the results on ImageNet are compared in a table.

Correctness: Yes/Yes

Clarity: Yes

Relation to Prior Work: Yes

Reproducibility: Yes

Additional Feedback: ***Rebuttal Response*** Thanks for the author's response. As all my concerns are addressed in the feedback, I will raise the score to accept the paper.

[Author Response · NeurIPS 2020]

We thank the reviewers for their feedback. We will reflect reviewer's comments and our response in the revision.

**Reviewers showed concern on the novelty and the accuracy.** FixMatch represents a significant research advancement *because* of its simplicity. While discovering new techniques and tricks is important, we believe that demonstrating state-of-the-art performance through the consolidation and simplification of existing concepts should be considered just as novel and important, if not more. Prior methods have implied that it is necessary to introduce complexity (for example, ReMixMatch uses self-supervised rotation losses, mixup, soft pseudo labels with sharpening, warmup, etc.) to achieve SOTA results. We find none of these are necessary; furthermore, they result in more hyper-parameters to tune which can be impractical on realistically-sized validation sets (as argued in [2]).

Another benefit of simplicity is the extensibility – for example, we showed that Distribution Alignment of ReMixMatch can be seamlessly added to FixMatch, improving an accuracy by 10% on CIFAR-100 with 400 labeled data.

**Hard versus Sharpened Soft labels (R2, R4):** We show through experiments that hard pseudo-label performs just as good or better than the sharpened soft pseudo-labels, with the desirable property of using one less hyperparameter.

**Barely-supervised learning (BSL) and few-shot learning (FSL) (R2):** We considered to compare BSL and FSL but found that they were not directly comparable. While they are similar in that they both use very small labeled training data, FSL is a subtask of transfer learning, where the model can be trained with additional "labeled" data from normal classes that are different from few-shot classes (e.g., [6], [4]), while BSL is not.

**Distribution Alignment (DA) (R4):** DA is more effective when the task is more challenging. Our experiments, which were done after paper submission, showed that DA is also important when training on ImageNet with 1% labeled data. On the other hand, we find DA effective as well when the amount of labeled data is small. For example, as in Section D.1, CIFAR-10 with 40 labeled examples also benefit from DA. We will add these results in the revision.

**How realistic the scenarios are (R1):** We refer the reviewer to Section 4.2. The unlabeled data of STL-10 is noisy: It contains samples from different classes than the 10 classes of the labeled set. Therefore it is more realistic and challenging. Thanks to confidence-based thresholding playing a role in rejecting out-of-distribution samples, FixMatch was able to achieve SOTA results in such scenarios without modification of parameters or algorithm design.

**Extension to other domains (R1):** There are application domains with advanced data augmentations, such as back-translation [5] for text classification or SpecAugment [3] for speech recognition, where FixMatch is readily applicable. Developing domain-agnostic SSL methods is desirable and a worthy long term research goal. For example, we have shown some potential of domain-agnostic FixMatch in Section D.2, showing improvements over other domain-agnostic SSL methods, though the absolute performance was not as good as with domain-specific data augmentation.

**Knowledge from large-labeled network training (R1):** It is true that lessons from supervised learning largely inspire the success of SSL. For imagenet experiments, we adopted a few such lessons, including learning schedule, network architecture, and data preprocessing and augmentations. While this may introduce unwanted inductive biases, FixMatch showed a solid improvement over previous works (e.g., UDA) under the same setting. Moreover, we also showed SOTA on STL-10, which is not widely used for large-scale supervised learning as it comes with only 5k labeled training data, by transferring an SSL setting from CIFAR-10.

**Weak and strong augmentation (R3):** Weak augmentation (e.g., shift and flip) provides a reliable prediction for pseudo-label generation. Strong augmentation prevents the consistency loss from being too easily minimized and the model from overfitting. We refer Section 5.2 on their importance where we conduct ablation studies comparing different combinations of weak/strong augmentations.

Strong augmentation may refer to a more diverse set of transformations and wider range of augmentation magnitudes. Though the data augmentation becomes an essential component for deep network training, less is known on how to quantify their impact [1], and the definition of weak and strong augmentations may vary across the dataset. We think that more systematic study on data augmentation would benefit FixMatch.

# References

[1] Gontijo-Lopes et al. Affinity and diversity: Quantifying mechanisms of data augmentation. *arXiv preprint arXiv:2002.08973*, 2020.

[2] A. Oliver et al. Realistic evaluation of deep semi-supervised learning algorithms. In *NeurIPS*, 2018.

[3] D. S. Park et al. Specaugment: A simple data augmentation method for automatic speech recognition. *arXiv preprint arXiv:1904.08779*, 2019.

[4] S. Ravi and H. Larochelle. Optimization as a model for few-shot learning. In *ICLR*, 2017.

[5] R. Sennrich et al. Improving neural machine translation models with monolingual data. *arXiv preprint arXiv:1511.06709*, 2015.

[6] O. Vinyals et al. Matching networks for one shot learning. In *NeurIPS*, 2016.


[Meta-Review · NeurIPS 2020]

Four knowledgeable reviewers support acceptance for the contributions. Reviewers find that i) the proposed algorithm is simple; ii) efficient and empirical evaluation is very carefully designed with an extensive ablation study; iii) analysis on augmentation strategy and sharpening also provide good insights. Therefore, I also recommend acceptance. However, please consider revising your paper to address all the concerns and comments from the reviewers.